# Targeting the Canonical WNT/β-Catenin Pathway in Cancer Treatment Using Non-Steroidal Anti-Inflammatory Drugs

**DOI:** 10.3390/cells8070726

**Published:** 2019-07-15

**Authors:** Alexandre Vallée, Yves Lecarpentier, Jean-Noël Vallée

**Affiliations:** 1Diagnosis and Therapeutic Center, Hypertension and Cardiovascular Prevention Unit, Hotel-Dieu Hospital, AP-HP, Université Paris Descartes, 75004 Paris, France; 2Centre de Recherche Clinique, Grand Hôpital de l’Est Francilien (GHEF), 6–8 rue Saint-fiacre, 77100 Meaux, France; 3Centre Hospitalier Universitaire (CHU) Amiens Picardie, Université Picardie Jules Verne (UPJV), 80054 Amiens, France; 4Laboratoire de Mathématiques et Applications (LMA), UMR CNRS 7348, Université de Poitiers, 86000 Poitiers, France

**Keywords:** non-steroidal anti-inflammatory drug, cancer, WNT, inflammation, oxidative stress, PPARγ

## Abstract

Chronic inflammation and oxidative stress are common and co-substantial pathological processes accompanying and contributing to cancers. Numerous epidemiological studies have indicated that non-steroidal anti-inflammatory drugs (NSAIDs) could have a positive effect on both the prevention of cancer and tumor therapy. Numerous hypotheses have postulated that NSAIDs could slow tumor growth by acting on both chronic inflammation and oxidative stress. This review takes a closer look at these hypotheses. In the cancer process, one of the major signaling pathways involved is the WNT/β-catenin pathway, which appears to be upregulated. This pathway is closely associated with both chronic inflammation and oxidative stress in cancers. The administration of NSAIDs has been observed to help in the downregulation of the WNT/β-catenin pathway and thus in the control of tumor growth. NSAIDs act as PPARγ agonists. The WNT/β-catenin pathway and PPARγ act in opposing manners. PPARγ agonists can promote cell cycle arrest, cell differentiation, and apoptosis, and can reduce inflammation, oxidative stress, proliferation, invasion, and cell migration. In parallel, the dysregulation of circadian rhythms (CRs) contributes to cancer development through the upregulation of the canonical WNT/β-catenin pathway. By stimulating PPARγ expression, NSAIDs can control CRs through the regulation of many key circadian genes. The administration of NSAIDs in cancer treatment would thus appear to be an interesting therapeutic strategy, which acts through their role in regulating WNT/β-catenin pathway and PPARγ activity levels.

## 1. Introduction

The complex process of cancer can be defined in terms of three stages: initiation, promotion, and progression [1,2,3]. Many cancers are initiated by chronic inflammation, involving numerous physical, chemical, and biological determinants [4]. Several studies have examined the association between chronic inflammation and cancer [5,6] and indeed have highlighted the promising role of anti-inflammatory treatments for cancer [7]. Chronic inflammation is responsible for the different stages observed in cancers, such as invasion, angiogenesis, proliferation, and metastasis [8,9,10].

In parallel, oxidative stress causes DNA damage in cancers [11]. In the past few years, the combined effect of oxidative stress and chronic inflammation has been the subject of several studies [12]. Reactive oxygen species production (ROS) is increased by the activation of inflammatory factors [13,14,15] and thus also participates in the processes of invasion, proliferation, angiogenesis, and then metastasis [16]. The canonical WNT/β-catenin pathway controls numerous other pathways involved in cancer development and tissue homeostasis. This pathway is regulated from transcription-level regulations to post-transcriptional modifications. In cancers, an aberrant WNT/β-catenin pathway is generally observed and leads to oxidative stress and inflammation [12,17,18]. 

Several epidemiological studies have shown that non-steroidal anti-inflammatory drugs (NSAIDs) could have a positive effect on both the prevention of cancer and tumor therapy. Moreover, the regular administration of aspirin, a NSAID, has been found to be correlated with a reduction in cancer incidence [19]. A regular therapy of more than 75 mg/day of aspirin diminishes the incidence of several cancers and tumor metastases, leading to an improvement in survival rates [20]. Regular use of NSAIDs is associated with a reduced incidence of several cancers, such as breast cancer, lung cancer, and gliomas [21,22]. Recent data have shown that the use of aspirin is associated with a reduction in the incidence of death from cancer, as well as in metastatic spread [19,20,23]. Anti-inflammatory drugs are commonly used in clinical practice due to their analgesic, anti-inflammatory and antipyretic effects. Furthermore, NSAIDs are often used in conjunction with other drugs in treating a number of diseases. Numerous hypotheses have postulated that NSAIDs could decrease tumor growth by acting on both chronic inflammation and oxidative stress [24]. Anti-inflammatory drugs could be used to target the chronic inflammatory microenvironment of tumors. It is well known that the human body is capable of self-healing after a short-term inflammatory response, but a long-term chronic inflammation could lead to initiation of the cancer process. Many studies have shown that inflammatory factors, including interleukins, TNF-α, nuclear factor-κB (NF-κB) and ROS production-induced inflammation, infiltrate the inflammatory microenvironment, leading to DNA damage and ultimately the initiation of the cancer process [25,26]. 

NSAIDS act as peroxisome proliferator-activated receptor gamma (PPARγ) agonists and could thus downregulate the aberrant WNT/β-catenin pathway in cancers [22]. PPARγ agonists offer an interesting therapeutic solution in cancers by acting on both oxidative stress and inflammation [27,28]. Indeed, in several tissues, canonical WNT/β-catenin pathway activation leads to inactivate PPARγ, while PPARγ activation inhibits the canonical WNT/β-catenin pathway. In cancers, the canonical WNT/β-catenin pathway is overactivated while PPARγ is decreased [12]. In parallel, the disruption of circadian rhythms (CRs) has been shown in cancers [29]. This dysregulation upregulates the canonical WNT/β-catenin pathway, which participates in the cancer process. PPARγ modulates CRs by regulating some circadian genes, such as Bmal1 (brain and muscle aryl-hydrocarbon receptor nuclear translocator-like 1) [30], and can directly target the WNT pathway [31]. Numerous evidence points to the anti-cancer benefits of NSAIDs, even if these benefits remain unclear and poorly understood. Nevertheless, data from experiments suggest a potential role for NSAIDs in the treatment of cancer through the regulation of the WNT/β-catenin pathway [32].

This review focuses on the interest of using NSAIDs in cancer therapy through their capacity to regulate the aberrant canonical WNT/β-catenin pathway and PPARγ, two systems that respond in an opposite manner.

## 2. Benefits and Disadvantages of NSAIDs-Based Cancer Prevention

### 2.1. NSAIDs and Cancer Prevention

A strategy for reducing cancer risks could involve the use of NSAIDs (such as aspirin, naproxen, or ibuprofen) [20]. Some studies have presented a correlation between the long-term use of aspirin and a reduction in both the incidence and mortality of cancers, a reduction that can vary from 20% to 75% [20]. The most marked effects have been observed in colorectal cancers, stomach cancers, and esophageal adenocarcinoma, while less marked effects have been found in lung, prostate, and breast cancers [20]. In contrast, little benefit has been observed in pancreatic and endometrial cancers [20]. Numerous molecular mechanisms could explain the link between NSAIDs and cancer prevention, such as COX inhibition, immune response, PI3K/Akt pathway downregulation, pro-inflammatory response, and decreased glycolytic signaling in tumor cells [22,33]. COX inhibition is associated with the reduction in inflammatory mediators including prostaglandins [34]. The activation of COX in the cancer process leads to the expression of prostaglandin E3 (PGE2), which induces angiogenesis, tumor growth and metastasis [35]. Furthermore, PGE2 stimulates several signaling pathways, such as the PI3K/Akt and NF-κB, which induce tumorigenesis [35]. Finally, recent studies have shown that NSAIDs could also act on other signaling pathways, such as iNOS, TNF-alpha and interleukins [33].

### 2.2. NSAIDs Lead to Cancer Cell Apoptosis

The use of NSAID (aspirin) in ovarian cancer cells decreases Bcl-2 expression and increases Bax gene expression [24]. The role of Bcl-2 in cancer is to inhibit apoptosis by changing mitochondria thiol, affecting mitochondria membrane permeability, and translocating to the mitochondria membrane the apoptotic protein precursor Apaf-1 to inhibit the role of the latter. The NSAID sulindac diminishes the expression of both the protein Bcl-XL and the Bcl-XL antagonist of Bax to induce activation of caspase cascade, which stimulates the apoptosis process [36]. Aspirin can change mitochondrial permeability to downregulate Bcl-2 expression, block ATP synthesis, and release cytochrome C, which triggers apoptosis [37]. The NSAID celecoxib activates the p53-upegulated modulator of apoptosis (PUMA) to increase p53 expression and thus initiate apoptosis [38]. 

### 2.3. NSAIDs Inhibit COX-2

NSAIDs are known to protect cells from one step on the path to cancer through the inhibition of COX [39]. COX presents three subtypes: COX-1, COX-2, and COX-3 (in the nervous system) [40,41]. COX-1 catalyzes the production of prostaglandins (PGs) to maintain physiological functions. COX-2, a membrane-bound protein, is not expressed in normal cells but overexpressed in inflammation and tumors [42]. The PG overexpression is induced by COX-2 catalysis during inflammation that leads to neovascularization to provide nutrition for tumor proliferation [43]. In parallel, in tumors, COX-2 upregulates Bcl-2 expression to initiate the anti-apoptotic process [44], while it modulates MMP-2 expression, which induces tumor invasion and metastasis [45]. Celecoxib, a NSAID and a COX-2 enzyme inhibitor, can downregulate tumor proliferation and can induce apoptosis in a variety of tumor cells [24]. By directly blocking COX-2 expression, NSAIDS could prevent cancer initiation [24].

### 2.4. NSAIDs and the Akt Pathway

Colorectal cancer presents epidermal growth factor receptor (EGFR) overexpression and the use of aspirin can downregulate EGFR [46]. EGFR is involved in several pathophysiological responses in cancer, such as migration, proliferation, and invasion [47,48]. The Akt pathway is stimulated by EGFR [49]. An aberrant WNT/β-catenin pathway activates EGFR activity [18]. Furthermore, NSAIDs dephosphorylate Akt signaling and decrease MMP-2 gene expression to stop invasion and cell growth [50,51].

### 2.5. NSAIDs and Their Side Effects

The long-term use of NSAIDs could result in the appearance of side effects such as renal failure and gastro-intestinal symptoms (bleeding, mucosal lesions, inflammation leading to intestinal strictures and perforation, peptic ulcers) [52]. The administration of NSAIDs also increases the risk of deep vein thrombosis and pulmonary embolism, myocardial infarction, and stroke [53,54,55]. Numerous COX inhibitors have been withdrawn because of the associated increased risk of thromboembolic events. Celecoxib remains the only selective COX inhibitor available in the US and Europe [56]. COX inhibitor herbal medicines, such as *Cordia myxa fruit*, would appear to be promising “NSAID-like” agents in that they inhibit cancer and inflammation [52]. 

### 2.6. PPARγ: A Therapeutic Solution Induced by NSAIDs

NSAIDS act as PPARγ agonists by inhibiting COX-2 in gliomas [57] and colon cancer [58]. In parallel, NSAIDs present a COX independent anti-carcinogenic action through the direct control of the PPARγ expression [59,60]. Several studies have shown that NSAID action is modulated by PPARγ [61,62]. Studies have shown the potential impact of NSAIDs through the interplay of PPARγ and the WNT/β-catenin pathway [22].

### 2.7. PPARγ in Cancers

The ligand-activated transcriptional factor peroxisome proliferator receptor γ (PPARγ) is a component of the nuclear hormone receptor super family. It makes a heterodimer with retinoid X receptor (RXR), forming a PPARγ-RXR complex that binds to specific peroxisome proliferator response element (PPRE) regions in the DNA. It also activates numerous target genes involved in fatty acid transport (FABP3), cholesterol metabolism (CYP7A1, LXRα, CYP27), glucose homeostasis (PEPCK, GyK), and lipid catabolism (SCD-1). This dimer acts on other coactivator proteins like PGC-1α, and leads to specific gene overexpressions [63]. Glucose homeostasis, insulin sensitivity, lipid metabolism, immune responses, cell fate, and inflammation are regulated by PPARγ activation [64,65]. Circadian variations in blood pressure and heart rate are controlled by PPARγ expression by its interaction on Bmal1 [30,66]. PPARγ controls the expression of numerous genes implicated in inflammation, and it diminishes the activity of inflammation-related transcription factors such as NF-κB [67]. Several studies have shown decreased PPARγ expression in association with chronic inflammation in cancers [12,68].

### 2.8. Benefits and Disadvantages of PPAR Gamma Agonists in Cancers

Some positive effects have been observed with the administration of PPAR gamma agonists in the cancer process. PPAR gamma stimulation could reduce cancer development by the arrest of cell proliferation and the inhibition of the tumor growth factor [69]. The decrease in cyclin D1, a WNT target, is associated with the downregulation of cyclin-dependent kinase (CDK) and thus attenuates the phosphorylation of the retinoblastoma (Rb) protein leading to the arrest of the cell cycle [69]. Moreover, PPAR gamma agonists could induce apoptosis through intrinsic and extrinsic apoptosis pathways [70]. PPAR gamma activation is associated with the diminution in anti-apoptotic proteins, including Bcl-2, and the increase in p53 and the Bcl-2-associated death promoter (BAD) protein (B-4) [71]. TNF pathway activity is decreased by PPAR gamma agonists, leading to apoptosis [72]. In pancreatic cancer, invasiveness is affected by PPAR gamma activation, leading to the improvement of MMP-2 and the expression of plasminogen activation inhibitor-1 [72]. PPAR gamma agonists inhibit VEGF, IL-8, COX and thus suspend tumor angiogenesis [73]. In addition, PPAR gamma agonists reduce glycolytic pathway activity by altering the nutrient pathway and WNT signaling [74]. 

However, the use of PPAR gamma agonists does have some side effects, even if new molecules now have fewer disadvantages [53]. Rosiglitazone has been correlated with an augmentation of myocardial ischemia [75], but results from other studies remain unclear, showing no significant increase in cardiovascular events [76]. PPAR gamma agonist therapies appear to be correlated with an augmented risk of heart failure [77]. Rosiglitazone therapy enhances the risk of fatal and non-fatal heart failure ([76], and similar results have been observed with pioglitazone therapy [78]. Weight gain, edema formation, and fluid retention are other side effects of PPAR gamma agonists [79]. The administration of PPAR gamma agonists could also be associated with increased vascular permeability, leading to the appearance of peripheral edema [80]. 

## 3. Two Major Mechanisms Involved in Cancers: Chronic Inflammation and Oxidative Stress

### 3.1. Chronic Inflammation in Cancers

Some studies have presented that chronic inflammation leads to DNA damage and tissue injury [81]. Chronic inflammation impairs cell homeostasis and metabolism initiating the development of cancer [82]. Moreover, DNA damage from chronic inflammation provides a point of origin for the development of malignancy sites [83,84]. The relationship between cancer and chronic inflammation has been well documented by numerous studies [12,85]. Chronic inflammation stimulates ROS and reactive nitrogen species (RNS) production, leading to DNA damage [86]. Thus, genomic instabilities are caused by DNA damage and lead to the cancer process [87]. Several sites of common pathogenic infections are related to cancer initiation [88,89].

The immune system is also regulated by several inflammatory factors, such as the tumor necrosis factor α (TNF-α), interleukin-6 (IL-6), vascular endothelial growth factor (VEGF), and tumor growth factor-β (TGF-β) [90,91]. TNF-α expression leads to DNA damage and cytokine stimulation (such as IL-17 [92]), which are responsible for tumor growth, invasion, and angiogenesis [93]. Interleukins IL6 and IL-17 stimulate the signal transducer and activator transcription (STAT) signaling involved in the cancer process [94].

Chronic inflammation is also responsible for an augmentation in cyclooxygenase 2 (COX-2, a prostaglandin-endoperoxidase synthase) [95]. Several cytokines (TNF-α, IL-1) activate COX-2 [90]. COX-2 stimulates ROS and RNS production [95,96]. NF-κB stimulates several pro-inflammatory factors that activate COX-2 and inducible nitric oxide synthase (iNOS) [82]. NF-κB is one of the major factors implicated in chronic inflammation in the cancer process [82,97]. Numerous studies have shown that NF-κB stimulates the expression of TNF-α, IL-6, IL-8, STAT3, COX-2, BCL-2 (B-cell lymphoma 2), metalloproteinases (MMPs), VEGF [82], and thus ROS production [98]. Il-6 and VEGF stimulate the STAT-3 pathway, which is involved in proliferation, angiogenesis and metastasis [99]. Numerous cancers show an over-activation of the STAT-3 pathway [100]. Furthermore, during chronic inflammation, iNOS, an enzyme catalyzing nitric oxide (NO), is stimulated [101], resulting in an increase in p53 gene mutations [90].

### 3.2. Oxidative Stress in Cancers

Oxidative stress is considered as an imbalance between the production and elimination of ROS and RNS [11,102]. Cell damage from oxidation and nitration of macromolecules enhances ROS production by activation of the NADPH oxidase (NOX) enzyme. This phenomenon leads to the reduction of the transfer of electrons through the mitochondrial membrane to reduce the molecular oxidative metabolism. ROS production has a key role in numerous signaling pathways that are involved in changes in the microenvironment [103]. Thus, dysfunctions in the respiratory chain of mitochondria are responsible for ROS production [104]. The inflammation observed in sites where there is damage involves the uptake of oxygen leading to the release of ROS and its accumulation [8,105]. NF-κB, STAT, hypoxia-inducible factors (HIF) and both activator protein-1 (AP-1) play a major role in stimulating this process [82]. Moreover, in a vicious circle, COX-2, TNF-α, IL-6, and iNOS are induced by oxidative stress [96]. NADPH-oxidase (NOX) is activated by chronic inflammation and increases oxidative stress, resulting in changes in nuclear signaling [106].

### 3.3. Interaction between Oxidative Stress and Inflammation

Numerous studies have shown that the phenomenon by which oxidative stress can enhance chronic inflammation, which in a negative feedback could lead to cancers [11] (Figure 1). The imbalance caused by oxidative stress leads to damage in the signaling in cells [102]. ROS have a main role both upstream and downstream of the NF-κB and TNF-α signaling pathways, which are the main mediators of the inflammation. The hydroxyl radical is the most harmful of all the ROS. A vicious circle is observed between ROS and these pathways. ROS are formed by the NOX system. Moreover, the proteins modified by ROS could result in initiation of the auto-immune response to stimulate TNF-α and thus NOX [107]. Nuclear factor erythroid-2 related factor 2 (Nrf2) is mainly associated with oxidative stress in inflammation [11]. Nrf2 is a transcription factor which binds with the antioxidant response element (ARE) [108]. The protective role of Nrf2 in cancer relates to its capacity to reduce inflammation and oxidative stress [109]. Several studies have presented that Nrf2 can have an anti-inflammatory role by regulating MAPK (mitogen-activated protein kinases), NF-κB, and PI3K pathways [110]. Thus, Nrf2 may play a major role in reducing oxidative damage [111]. Evidence suggests that mitochondrial dysregulation has a significant role in the cancer process [11].

## 4. The WNT/β-Catenin Pathway

### 4.1. The WNT Pathway, Chronic Inflammation, and Oxidative Stress

Many studies have observed that the canonical WNT/β-catenin pathway increases the inflammatory process [81]. Furthermore, infection pathogens activate the WNT/β-catenin pathway, thereby enhancing inflammation [112]. ROS, stimulated by NOX, activates the canonical WNT/β-catenin pathway through the oxidization and inhibition of the nucleoredoxin (a redox-sensitive regulator), thus stimulating the cancer process [82]. ROS production stimulates c-Myc [113], STAT [114] and phosphatidylinositol-3-kinase (PI3K/Akt) [115], and the inhibition of PPARγ [116]. ROS production stimulates Akt signaling by inhibiting the phosphatase and tensin homolog deleted from chromosome (PTEN) [117,118]. The canonical WNT/β-catenin pathway may thus have a key role in the cancer process by stimulating both oxidative stress and inflammation [12]. The WNT pathway is the target of several inhibitors in therapeutic strategies to counteract tumorigenesis, such as OMP-54F28 [119,120,121], frizzled antibodies [122], tankyrase inhibitors [123], CBP inhibitors [124], and PORCN inhibitors [125].

### 4.2. The Sanonical WNT/β-Catenin Pathway, a Major Factor in Cancer Development

The name WNT is derived from wingless drosophila melanogaster and its mouse homolog Int. The WNT pathway is implicated in several pathways and regulating signaling pathways, like embryogenesis, cell proliferation, migration and polarity, apoptosis, and organogenesis [126]. During the adult stage, the WNT pathway is non-activated or silent. However, in numerous mechanisms and pathologies, such as inflammatory, metabolic and neurological disorders, and cancers, the WNT pathway may become dysregulated [127]. Some studies have utilized the WNT pathway for the cell therapy-bioengineering processes [128].

WNT ligands are lipoproteins that stimulate specific co-receptors (Figure 2). These WNT ligands stimulate the canonical WNT pathway by activation of β-catenin. WNT ligands stimulate frizzled (FZD) receptors and low-density lipoprotein receptor-related protein 5 and 6 (LRP 5/6) [129,130]. The complex formed by these extracellular WNT ligands and FZD/LRP5/6 activates the intracellular disheveled (DSH). This stimulation leads to inactivate the destruction complex of β-catenin in the cytoplasm. B-catenin accumulates in the cytoplasm and then migrates into the nucleus. Nuclear β-catenin links to T-Cell factor/lymphoid enhancer factor (TCF/LEF) to activate target gene transcriptors, such as c-Myc and cyclin D1 [131].

During the “off-state” of the WNT/β-catenin pathway, WNT ligands do not interact with FZD and LRP 5/6. The β-catenin destruction complex, composed by AXIN, APC (adenomatous polyposis coli) and GSK-3β (glycogen synthase kinase 3β), phosphorylates β-catenin. Then, phosphorylated β-catenin is destroyed into the proteasome.

Many WNT inhibitors downregulate the canonical WNT/β-catenin pathway. GSK-3β is the main inhibitor of the WNT pathway. GSK-3β is a neuron-specific intracellular serine-threonine kinase that controls numerous signaling pathways like inflammation, neuronal polarity, and cell membrane signaling [132,133,134]. GSK-3β downregulates β-catenin cytosolic stabilization and nuclear migration. Dickkopf (DKK) and soluble frizzled-related proteins (SFRP) are also WNT inhibitors and interact with FZD, LRP5, and LRP6 [135,136,137].

### 4.3. WNT and Inflammation in Cancer

Positive interplay between WNT/β-catenin and NF-κB has been observed [138]. The activation of the WNT/β-catenin leads to stimulate IκB-α (nuclear factor of kappa light polypeptide gene enhancer in B-cells inhibitor, alpha) degradation and then NF-κB stimulation [139]. Stimulation of the target gene, CRD-BP (coding region determinant-binding protein, an RNA-binding protein), by activated β-catenin stabilizes mRNA of βTrCP (B-transducin repeat-containing protein) [140]. In colon cancer, the activation of both βTrCP and CRD-BP is correlated with the stimulation of the β-catenin and NF-κB, leading to proliferation and metastasis [140,141]. In breast cancer, TLR3 activation stimulates β-catenin, leading to overstimulation of the NF-κB pathway [142]. Moreover, the β-catenin and NF-κB pathways stimulate each other in diffuse large B-cell lymphomas [143]. The WNT/β-catenin pathway activates COX-2, which then enhances the inflammatory response [144]. E-cadherin and GSK-3β are downregulated in melanoma cells by β-catenin pathway [145]. Concomitant GSK-3β and E-cadherin inhibition with cytoplasmic accumulation of β-catenin leads to NF-κB-dependent iNOS expression in hepatic cells [146]. The WNT/β-catenin signaling stimulates its target TNFRSF19 in colon cancer, that leads to activation of NF-κB signaling [147]. Nevertheless, the observed synergistic interaction between β-catenin and NF-κB depends on the β-catenin-TCF/LEF link [148].

NF-κB overexpression inactivates GSK-3β whereas it stimulates β-catenin signaling [149,150]. GSK-3β activation leads to the downregulation of TNF-α-induced NF-κB activation in carcinoma cells [149]. IκB is stabilized by GSK-3β activation, resulting in the downregulation of the NF-κB pathway [150]. NF-κB signaling can modulate the WNT/β-catenin pathway by IKKα (IκB Kinase-α) use [151] and RelA [152]. IKKα stimulates β-catenin signaling while IKKβ inhibits β-catenin pathway [153]. IKKα activates the β-catenin/TCF/LEF link [154]. The stimulation of IKKα leads to the cytoplasmic accumulation of β-catenin resulting in GSK3-β and APC inactivation [151].

### 4.4. WNT and Oxidative Stress in Cancer

The over-activated PI3K/Akt pathway observed in the cancer process is stimulated by ROS production [155,156]. PTEN is the main inhibitor of the PI3K/Akt pathway [118]. NADPH oxidase and superoxide dismutase oxidize PTEN to inhibit it. Downregulation of PTEN leads to an increase in Akt activity, which enhances the phosphorylation of GSK-3β. Thus, GSK-3β inhibited by Akt does not bind β-catenin. Inactivation of PTEN stimulates Akt and β-catenin [157]. Moreover, ROS production participates in the stabilization of HIF-1α thereby activating glycolytic enzymes [68,155]. The WNT/β-catenin pathway stimulates HIF-1α by activating the PI3K/Akt pathway [18]. Although this mechanism remains unclear, recent studies have presented that ROS production stimulates the WNT/β-catenin pathway [158]. In parallel, Akt [159] and c-Myc [160] enhance ROS production.

## 5. Interplay between NSAIDs - PPARγ and the WNT/β-Catenin Pathway in Cancers

### 5.1. PPAR Gamma and the WNT/β-Catenin Pathway

The role of PPARγ agonists remains unclear in cancer cells, even if their role is well understood in the regulation of differentiation and stemness programs [161]. In physiological cells, PPARγ inhibits tumorigenesis and WNT signaling through the target of phosphorylated β-catenin at the proteasome through a process that involves its catenin-binding domain within PPARγ. Nevertheless, oncogenic β-catenin counteracts proteasomal degradation by downregulating PPARγ expression, that needs its TCF/LEF binding domain [162]. In adipocyte cells, PPARγ leads to increased differentiation and a reduction in proliferation by targeting the WNT/β-catenin pathway. PPARγ binds with GSK3-β to activate the differentiation factor C/EBPα, leading to adiponectin production [74,163]. PPARγ stimulation downregulates β-catenin at both the mRNA and protein levels to induce differentiation [164]. In metastatic prostate cancer LnCaP cells, PPARγ inhibits the WNT pathway by affecting phosphorylated β-catenin in the proteasome [162,165]. In colorectal and gastric cancer cells, PPARγ inhibits β-catenin signaling, cytoplasmic localization and target effectors, leading to the control of numerous genes, including telomerase reverse transcriptase and Sox9, both of which are implicated in cell differentiation and the survival phenomenon [166,167,168]. PPARγ agonists, by decreasing the WNT/β-catenin pathway, may be utilized in association with other drugs, including inhibitors of tyrosine kinases [169], Akt [170], and MAPK cascades, to maximize the anti-tumor and pro-differentiating effect.

### 5.2. NSAIDs and the WNT/β-Catenin Pathway

NSAIDs downregulate the activity of COX-2 and thereby inhibit the synthesis of prostaglandins (PGE2) and then the WNT pathway [171]. Recent studies have observed that NSAIDS can have an anti-tumor effect, having revealed a chemo preventive effect against colon cancer [172,173,174]. The possible cellular pathway underlying the chemo preventive effect of NSAIDs involves the induction of cell-cycle arrest, apoptosis, and angiogenesis inhibition [173,174].

Several studies have also shown that NSAIDs can inhibit the canonical WNT/β-catenin pathway [22]. Both aspirin and indomethacin downregulate the transcriptional activity of β-catenin/TCF-responsive genes [175]. NSAIDs diminishes nuclear β-catenin levels and leads to the degradation of β-catenin [176]. The NSAIDs, including sulindac, exisulind, and celecoxib, inhibit β-catenin levels and then decrease the transcriptional activity of the β-catenin/TCF/LEF complex [177]. Celecoxib directly decreases cancer cell growth by downregulating the expression of the WNT/β-catenin signaling pathway [178] and by inducing the degradation of the TCF7L2 [179]. Sulindac also inhibits the WNT/β-catenin pathway by downregulating nuclear β-catenin localization and β-catenin/TCF target gene transcription [180]. Colon cancer therapy with celecoxib is associated with an inhibition of the canonical WNT/β-catenin pathway [181,182]. Celecoxib inhibits the activity of the complex TCF/LEF and thus the activity of cyclin D1, suggesting that this component inhibits the expression of WNT/β-catenin target genes [182]. Aspirin decreases glioma cell proliferation and invasion by inhibiting β-catenin/TCF transcription [183]. It also stops glioma cells cycle at the G0/G1 phase and inhibits invasion and tumor growth by downregulating β-catenin/TCF activity [183,184].

## 6. Circadian clock: An Interesting Pathway in Cancer Development

### 6.1. Circadian Clock

Numerous biological phenomena in the body are regulated by the circadian “clock” (circadian locomotors output cycles kaput). The circadian clock is in the hypothalamic suprachiasmatic nucleus (SCN). CRs are endogenous and entrainable free-running periods that last approximately 24 h. Numerous transcription factors can regulate CRs. These are called circadian locomotor output cycles kaput (Clock), brain and muscle aryl-hydrocarbon receptor nuclear translocator-like 1 (Bmal1), Period 1 (Per1), Period 2 (Per2), Period 3 (Per3), and Cryptochrome (Cry 1 and Cry 2) [185,186] (Figure 3). These factors are controlled by positive and negative self-regulation mediated by CRs [187,188]. Clock and Bmal1 heterodimerize and thus enhance the transcription of Per1, Per2, Cry1, and Cry2 [189]. The Per/Cry heterodimer can downregulate its stimulation through negative feedback. It migrates back to the nucleus to directly inhibit the Clock/Bmal1 complex and then repress its own transcription [189]. The Clock/Bmal1 heterodimer also stimulates the transcription of retinoic acid-related orphan nuclear receptors, Rev-Erbs, and retinoid-related orphan receptors (RORs). By a positive feedback, RORs stimulate Bmal1 transcription, while Rev-Erbs inhibit their transcription by a negative feedback [189].

### 6.2. Circadian Clock Disruption in Cancers

Epidemiological and fundamental evidence supports the idea of linking circadian disruption with cancer [29]. DNA repair, apoptosis and cell cycle regulation follow circadian rhythms in humans [190]. Disruption of the CRs is correlated with dysregulation in cell proliferation and thus the initiation of cancer [191]. Clock/Bmal1, Per1 and Per2 maintain the rhythmic pattern of cell proliferation and repair of DNA damage [192,193]. Bmal1 overexpression has been observed in cell growth of NIH 3T3 cells [194]. Metastatic cancers present high levels of Clock or Bmal1 genes [195,196]. Clock overexpression is often associated with cell proliferation in colorectal carcinoma cells [197]. Bmal1 upregulation is found in certain types of pleural mesothelioma while Bmal1 knockdown is associated with reduced cell growth and induced apoptosis [198]. Bmal1 is considered an attractive target in leukemia cells [199].

## 7. Circadian Clock Disruption Enhances Both Inflammation and Oxidative Stress

### 7.1. Circadian Clock and Inflammation

Melatonin has been used in the treatment of chronic bowel inflammation resulting in decreasing inflammation through inhibition of COX-2 and iNOS [200]. Moreover, melatonin can act on iNOS and COX-2 by inhibiting p52 acetylation and transactivation [201]. Melatonin inhibits NF-κB and COX-2 in murine macrophage-like cells [202]. An anti-inflammatory response of melatonin has been observed through a decrease in NF-κB activity [203]. Melatonin downregulates the nuclear translocation of NF-κB, leading to an enhancement of anti-cancer effects in lung cancer [204].

### 7.2. Circadian Clock and Oxidative Stress

Recent studies have indicated that the hypoxic response in cancer could be directly controlled by the circadian rhythm Clock/Bmal1 [205]. In a similar way, blood oxygen levels present daily rhythms influenced by clock genes [206]. Metabolic dysregulation in cancers may result in disruption of Bmal1 in a hypoxic-dependent way [207]. Considerable evidence connects circadian disruption with hormone-dependent diseases, like breast and prostate cancers. One of the main factors is melatonin, a hormone produced by the pineal gland in a circadian manner to control sleep [208]. In the mitochondria, melatonin is linked to the regulation of oxidative stress [209]. Melatonin increases glutathione peroxidase and glutathione reductase activities [210]. Moreover, melatonin directly regulates the mitochondrial respiratory chain, which modulates ATP production [209]. Furthermore, alteration of melatonin secretion by sleep disruption could enhance ROS and RNS production [211].

## 8. WNT, NSAIDs, and PPAR Gamma with Circadian Clocks

### 8.1. The WNT/β-Catenin Pathway and the Circadian Clock

The WNT/β-catenin pathway is the downstream target of the RORs control factors and has several putative Bmal1 clock-binding sites within its promoter [212] (Figure 4). Through such relationships, circadian genes can control the progression of the cell cycle by the WNT signaling [213]. The WNT pathway can be inhibited by a Bmal1 knockdown [214]. Levels of WNT-related genes in wild-type mice are higher than those observed in Bmal1 knockdown mice [215,216]. Cell proliferation and cell cycle progression are controlled by Bmal1 through the activation of the canonical WNT/β-catenin pathway [217]. Bmal1 enhances β-catenin transcription, inhibits the degradation of β-catenin and downregulates GSK-3β activity [218]. Per2 degradation induced by β-catenin increases circadian disruption in the intestinal mucosa of ApcMin/+ mice [219].

In physiological circumstances, the core circadian genes work in accurate feedback circles and keep the molecular clockworks in the SCN. They permit the regulation of peripheral clocks [187,188]. Per1 and Per2 maintain cell circadian rhythms and control cell-related gene activity, including c-Myc, so as to sustain the physiologic cell cycle [220,221]. mRNAs and proteins levels of circadian genes oscillate throughout the 24-hour period [187].

### 8.2. NSAIDs and the Circadian Clock

Few studies have investigated the role of NSAIDs regarding the circadian clock. Nevertheless, Kowanko et al. observed that pain reported in rheumatoid arthritis after a twice a day therapy by flurbiprofen may be more efficient than four times daily flurbiprogen, and that regimen without an evening dose was the least efficient of three twice-daily therapies tested. Moreover, their results showed that morning stiffness in rheumatoid arthritis was not only the result of nocturnal inactivity but also a response to an appropriately timed medication [222]. Moreover, patients with indomethacin and ketoprofen have shown a reduction in neurological and gastro-intestinal side effects when these products are ingested once-daily in the evening rather than in the morning [223].

### 8.3. PPARγ and the Circadian Clock

PPARγ acts directly with the core clock genes and presents diurnal variations in liver and blood vessels [30,224] (Figure 4). In mice, dysregulation in diurnal rhythms are induced by the inhibition of PPARγ [225]. PPARγ agonists regulate Bmal1 and thus the heterodimer Clock/Bmal1 formation [30,226], and can interact with Rev-Erb [227]. Downregulation of the clock-controlled gene Nocturin inhibits PPARγ oscillations in the liver of mice fed on a high-fat diet. In physiological circumstances, nocturin interacts with PPARγ to enhance its transcriptional activity [228]. PPARγ diminution alters the circadian function of 15-Deoxy-D 12,14-prostaglandin J2 (15-PGJ2) [225]. The associate of PPARγ, RXR, interacts with Clock protein in a ligand-dependent manner and then blocks Clock/Bmal1 heterodimer formation and transcriptional activity [229]. PPARγ binds to the mammalian clock to control metabolic metabolism [229]. Circadian metabolism is controlled by PPARγ in a direct manner [225]. Retinoic acid receptor-related orphan receptor gamma t (ROR gammat) is considered a major transcriptional factor for Th17 cell differentiation [230,231]. Th17 cells represent another subset of CD4+ T cells and selectively produce interleukin (IL)-17. PPARγ can influence the function of Th cell clones [232]. PPARγ agonists decrease Th17 differentiation by inhibiting ROR gammat induction [233,234,235]. CD4+ T cells fail to express ROR gammat under the action of PPARγ agonists [233].

## 9. Relevance of “Chronotherapy” in Cancer Clinical Therapy

The numerous interactions between clock dysregulation and cancer underline the interest of circadian therapeutic actions [29]. The temporal peak of cell activity could be targeted by pharmacological drugs used at an optimal time of day. Few studies have shown the potential role of WNT and PPAR gamma with circadian clocks in cancer development. Nevertheless, interest in the association between PPAR gamma agonists and melatonin in cancer therapy is not new [236]. In cultured cells, the addition of melatonin with a PPAR gamma agonist (such as troglitazone) is associated with a significant reduction in cell numbers [237]. Moreover, other studies have shown a potent apoptotic effect of a combination of melatonin with PPAR gamma agonists in breast cancer cells [238,239]. In parallel, recent studies have shown that melatonin could inhibit WNT pathway expression [240,241].

In mouse ovaries, melatonin administration protects against ROS production and mitochondrial damage [242]. In colorectal cancer, the combination of 5-fluorouracil and melatonin is correlated with the downregulation of cell proliferation by suppressing of the PI3K/Akt pathway, NF-κB pathway and nitric oxide synthase signaling [243]. Moreover, melatonin inhibits GSK3-β to stop invasion in breast cancer cells [244]. The association between carcinogenesis and the circadian clock remains complex and difficult to unravel. Strong evidence shows the involvement of the circadian clock in cancer development. Numerous molecular pathways are dynamically circadian, such as the WNT/β-catenin pathway and PPAR gamma. Thus, the time at which these pathways are targeted may be critical. NSAIDs, by acting as PPAR gamma agonists and focusing on the WNT/β-catenin pathway, should be utilized in concordance with the circadian clock genes, and therefore administered at the optimum time of day. Further studies should focus on the importance of the day/night cycle in cancer therapy and the circadian profiles of cancer cells.

## 10. Conclusions

Chronic inflammation, oxidative stress and the disruption of circadian rhythms are important factors in the cancer process and are enhanced by overstimulation of the WNT/β-catenin pathway. In cancers, the WNT/β-catenin pathway is generally activated whereas PPARγ is decreased. These two signaling pathways act in opposing manners and this could explain their unidirectional profile observed in cancers. The use of NSAIDs, which act as PPARγ agonists, could be interesting in the reduction of both chronic inflammation and oxidative stress and in the control of circadian rhythms by inhibiting the WNT/β-catenin pathway (Figure 5). Due to the considerable impact of cancers on mortality and morbidity rates worldwide, it would appear of the utmost importance to better understand the action of NSAIDs in cancers (Table 1), and particularly their role in the inhibition of the major signaling system known as the WNT/β-catenin pathway.

## Figures and Tables

**Figure 1 cells-08-00726-f001:**
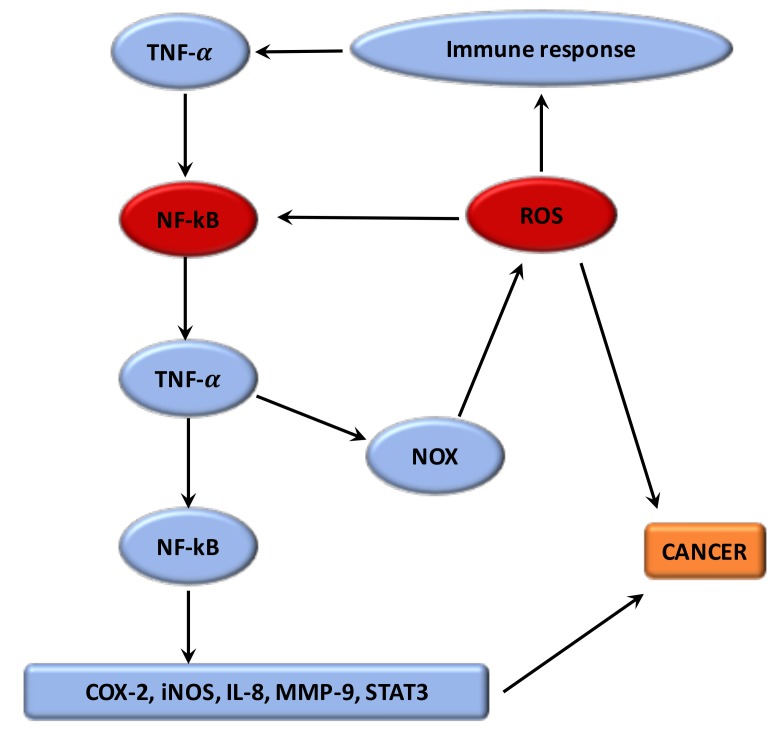
**Relationship between ROS and chronic inflammation.** The imbalance caused by oxidative stress leads to damage in the signaling in cells. ROS have a key role both upstream and downstream of the NF-κB and TNF-α signaling pathways, which are the main mediators of the inflammation. A vicious circle is observed between ROS and these pathways. ROS are generated by the NOX system. Proteins modified by ROS could result in activation of the auto-immune response to stimulate TNF-α and thus NOX. The dysregulation of these targets leads to the activation of several pathways involved in cancer initiation.

**Figure 2 cells-08-00726-f002:**
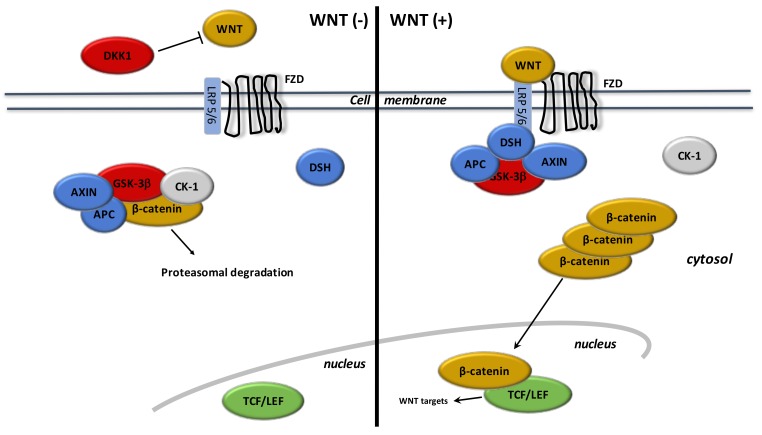
**The canonical WNT/β-catenin pathway. WNT (−).** Under physiologic circumstances, the cytoplasmic β-catenin is linked to its destruction complex, consisting of APC, AXIN, and GSK-3β. After CK-1 phosphorylates on Ser45 residue, β-catenin is phosphorylated on Thr41, Ser37, and Ser33 residues by GSK-3β. Thus, phosphorylated β-catenin is destroyed into the proteasome. Then, cytoplasmic level of β-catenin is kept low in the non-presence of WNT ligands. If β-catenin is not accumulated in the nucleus, the TCF/LEF complex does not stimulate the target genes. DKK1 inhibits the WNT/β-catenin pathway through the bind to WNT ligands or LRP5/6. **WNT (+).** When WNT ligands activate both FZD and LRP5/6, DSH is stimulated and phosphorylated by FZD. Phosphorylated DSH in turn activates AXIN, which comes off β-catenin destruction complex. Thus, β-catenin escapes from phosphorylation and then accumulates in the cytoplasm. The accumulated cytosolic β-catenin moves into the nucleus, where it interacts with TCF/LEF and stimulates the transcription of target genes.

**Figure 3 cells-08-00726-f003:**
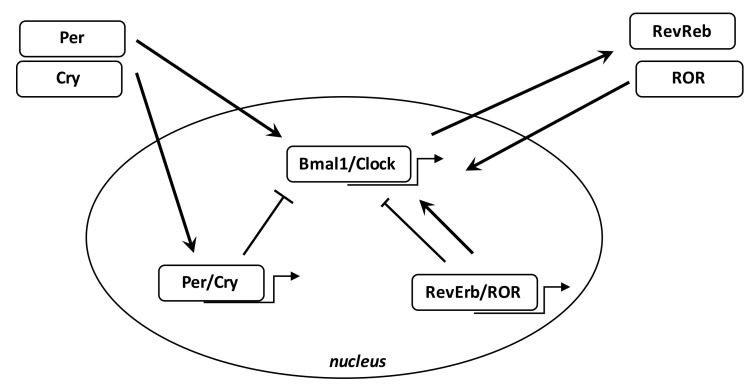
**Circadian clock genes.** The clock consists of a stimulatory circle, with the Bmal1/Clock heterodimer activating the transcription of Per and Cry genes, and an inhibitory feedback circle with the Per/Cry heterodimer translocating to the nucleus and repressing the transcription of the Clock and Bmal1 genes. An additional circle involves the RORs and RevErbs factors with a positive feedback by ROR and a negative feedback by RevErbs.

**Figure 4 cells-08-00726-f004:**
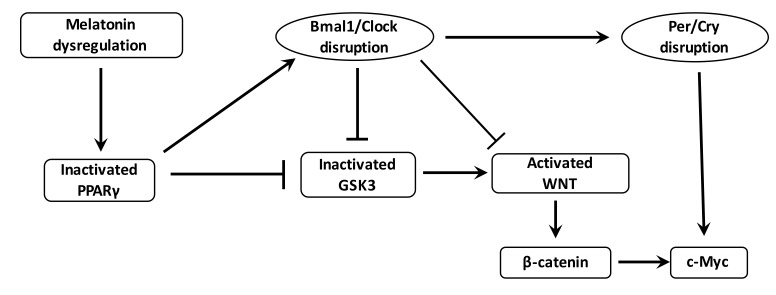
**Interactions between PPARγ, WNT pathway and circadian rhythms in cancer.** Dysregulation of melatonin and nocturin decreases the expression of PPARγ in cancer. Decreased PPARγ dysregulates the Bmal1/Clock heterodimer. Decreased PPARγ expression directly activates the formation of the heterodimer Bmal1/Clock and β-catenin cytosolic accumulation but inhibits the activity of GSK3, the main inhibitor of the WNT/β-catenin pathway. Bmal1/Clock knockout also decreases GSK3 activity and activates the WNT/β-catenin pathway and its downstream gene c-Myc through the stimulation of the heterodimer Per/Cry. The activation of the WNT/β-catenin pathway by the cytosolic accumulation of the β-catenin and the activation of c-Myc lead to cancer initiation (oxidative stress and chronic inflammation).

**Figure 5 cells-08-00726-f005:**
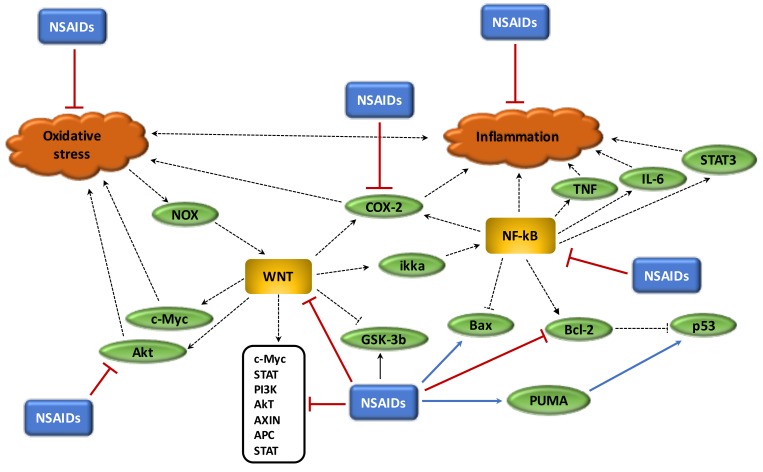
**Beneficial role of NSAIDs in cancer process.** (1) NSAIDs reduce oxidative stress; (2) NSAIDs reduce chronic inflammation; (3) NSAIDs inhibit Akt pathway activity; (4) NSAIDs downregulate WNT pathway and its target genes, inhibit Bcl-2, activate PUMA to stimulate p53 and activate GSK-3beta; (5) NSAIDs inhibit NF-κB and COX-2.

**Table 1 cells-08-00726-t001:** Differential effects of NSAIDs in tumors.

NSAIDs	Target	Target Function	Interaction	References
Aspirin	PGE2	Immune system attenuation	Inhibition of PGE2 synthesis	[245]
Aspirin	Platelets	Reduction cell activity	Inhibition of COX	[246]
Aspirin	Genetic mutations	Tumorigenesis inhibition	Downregulation of gene mutation accumulation	[247]
Aspirin	WNT pathway	Inhibition of cell proliferation and invasion	Inhibition of β-catenin accumulation	[183]
Aspirin	WNT pathway	Tumor suppressor	Inhibition of COX	[39]
Aspirin	WNT pathway	Arrest G0/G1 phase	Inhibition of β-catenin/TCF	[183,184]
Indomethacin	T-cell therapy	Tumorigenesis inhibition	Downregulation of cellular drug resistance	[246]
Sulindac	WNT pathway	Inhibition of invasion and cell growth	Phosphorylation of Akt signaling	[50,51]
Aspirin and indomethacin	WNT pathway	Tumorigenesis inhibition	Inhibition of β-catenin and TCF/LEF	[175]
celecoxib	WNT pathway	Inhibition cancer cell growth	Inhibition of WNT	[178]
celecoxib	WNT pathway	Inhibition cancer cell growth	Inhibition of TCF/LEF	[179]
Sulindac	WNT pathway	Tumorigenesis inhibition	Inhibition of β-catenin accumulation	[180]
celecoxib	WNT pathway	Tumorigenesis inhibition	Inhibition of cyclin D1	[181,182]

PGE2: prostaglandin E2, COX: cyclooxygenase, TCF/LEF: T-cell factor/lymphoid enhancer-binding factor.

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
