# Peer review of "Targeting the Canonical WNT/β-Catenin Pathway in Cancer Treatment Using Non-Steroidal Anti-Inflammatory Drugs"

_cells, 2019, doi:10.3390/cells8070726_

Round 1

Reviewer 1 Report

A review of the potential interactions between the WNT/ β-catenin pathway, PPARγ and circadian clocks should be of interest to many readers. However, this review was not as informative as expected. Information is provided about each individual pathway and what is known, but little analysis of the current state of the field, strengths and weaknesses, or future directions was provided. In addition, numerous grammatical errors are located throughout the manuscript. The first 3-4 pages of the manuscript (and abstract) are especially poorly written; errors and imprecise words make these sections almost unreadable. The review is also poorly organized (some paragraphs are only 1-2 sentences) and redundant.

Other points:

Current controversies regarding the use of NSAIDs for cancer prevention are not discussed. Data is stronger for some cancers than others so specific recommendations should be included. Similarly, the WNT/β-catenin pathway is most relevant to a few specific types of cancer. PPARγ is also a controversial target, which should be explained. What are the possible limitations of targeting these pathways?

The legend for Figure 1 needs to be expanded to explain the figure, independent of the text.

Are all of the effects of NSAIDs direct effects? Are any indirect effects, and if so, which pathways?

References should also be updated, as the majority of the references are more than 5 years old.

Author Response

The authors thank the Reviewer for his criticisms and suggestions, which have improved the form and substance of our manuscript. We hope to have answered all the questions asked. 

The additions and changes requested are written in red in the manuscript.

Comments and Suggestions for Authors

A review of the potential interactions between the WNT/ β-catenin pathway, PPARγand circadian clocks should be of interest to many readers. However, this review was not as informative as expected. Information is provided about each individual pathway and what is known, but little analysis of the current state of the field, strengths and weaknesses, or future directions was provided. In addition, numerous grammatical errors are located throughout the manuscript. The first 3-4 pages of the manuscript (and abstract) are especially poorly written; errors and imprecise words make these sections almost unreadable. The review is also poorly organized (some paragraphs are only 1-2 sentences) and redundant. 

Other points:

Current controversies regarding the use of NSAIDs for cancer prevention are not discussed. 

The observation of the reviewer is very relevant. We have added a paragraph:

Line 173

NSAIDs and their side effects

The long-term use of NSAIDs could result in the appearance of side effects such as renal failure and gastro-intestinal symptoms (bleeding, mucosal lesions, inflammation leading to intestinal strictures and perforation, peptic ulcers) [53]. The administration of NSAIDs also increases the risk of deep vein thrombosis and pulmonary embolism, myocardial infarction and stroke [54–56]. Numerous COX inhibitors have been withdrawn because of the associated increased risk of thromboembolic events. Celecoxib remains the only selective COX inhibitor available in the US and Europe [57]. COX inhibitor herbal medicines, such as Cordia myxa fruit, would appear to be promising “NSAID-like” agents in that they inhibit cancer and inflammation [53].”

Data is stronger for some cancers than others so specific recommendations should be included. Similarly, the WNT/β-catenin pathway is most relevant to a few specific types of cancer. PPARγis also a controversial target, which should be explained. What are the possible limitations of targeting these pathways? 

The reviewer is right. According to his remark, we have added more information about this signaling pathway

Line 183

“PPARγ: a therapeutic solution induced by NSAIDs

NSAIDS act as PPARγ agonists by inhibiting COX-2 in gliomas [58]and colon cancer [59]. In parallel, NSAIDs present a COX independent anti-carcinogenic action through the direct control of PPARγ expression [60,61]. Several studies have shown that NSAID action is modulated by PPARγ[62,63]. Recent studies have shown the potential impact of NSAIDs through the interplay of PPARγ and the WNT/β-catenin pathway [22].

PPARγ in cancers

The ligand-activated transcriptional factor peroxisome proliferator receptor γ (PPARγ) is a member of the nuclear hormone receptor super family. It forms a heterodimer with retinoid X receptor (RXR), leading to a PPARγ-RXR complex that binds to specific peroxisome proliferator response element (PPRE) regions in the DNA. It also activates several target genes involved in fatty acid transport (FABP3), cholesterol metabolism (CYP7A1, LXRα, CYP27), glucose homeostasis (PEPCK, GyK) and lipid catabolism (SCD-1). This dimer interacts with other coactivator proteins such as PGC-1α, and induces specific gene expressions [64]. Glucose homeostasis, insulin sensitivity, lipid metabolism, immune responses, cell fate and inflammation are regulated by PPARγ activation [65,66]. Circadian variations in blood pressure and heart rate are regulated by PPARγ through its action on Bmal1 [30,67]. PPARγ modulates the expression of several genes involved in inflammation, and it decreases the activity of inflammation-related transcription factors such as NF-ϰB [68]. Several studies have shown decreased PPARγ expression in association with chronic inflammation in cancers [12,69].

Benefits and disadvantages of PPAR gamma agonists in cancers

Some positive effects have been observed with the administration of PPAR gamma agonists in the cancer process. PPAR gamma stimulation could reduce cancer development by the arrest of cell proliferation and the inhibition of the tumor growth factor [70]. The decrease in cyclin D1, a WNT target, is associated with downregulation of cyclin-dependent kinase (CDK) and thus attenuates the phosphorylation of retinoblastoma (Rb) protein leading to the arrest of the cell cycle [70]. Moreover, PPAR gamma agonists could induce apoptosis through intrinsic and extrinsic apoptosis pathways [71]. PPAR gamma activation is associated with the decrease in anti-apoptotic proteins, including Bcl-2, and the increase in p53 and the Bcl-2-associated death promoter (BAD) protein (B-4)[72]. TNF pathway activity is decreased by PPAR gamma agonists, leading to apoptosis [73]. In pancreatic cancer, invasiveness is affected by PPAR gamma activation, leading to the improvement of MMP-2 and the expression of plasminogen activation inhibitor-1 [73]. PPAR gamma agonists inhibit VEGF, IL-8, COX and thus suspend tumor angiogenesis [74]. In addition, PPAR gamma agonists reduce glycolytic pathway activity by altering the nutrient pathway and WNT signaling [75]

However, the use of PPAR gamma agonists does have some side effects, even if new molecules now have fewer disadvantages [54]. Rosiglitazone has been associated with an increase of myocardial ischemia [76], but results from other studies remain unclear, showing no significant increase in cardiovascular events [77]. PPAR gamma agonist therapies appear to be correlated with an increased risk of heart failure [78]. Rosiglitazone therapy increases the risk of fatal and non-fatal heart failure ([77], and similar results have been observed with pioglitazone therapy [79]. Weight gain, edema formation and fluid retention are other side effects of PPAR gamma agonists [80]. The administration of PPAR gamma agonists could also be associated with increased vascular permeability, leading to the appearance of peripheral edema [81].”

The legend for Figure 1 needs to be expanded to explain the figure, independent of the text.

The reviewer is right. We have added a legend to the figure 1:

“Figure 1: Relationship between ROS and chronic inflammation 

The imbalance caused by oxidative stress leads to damage in the signaling in cells. ROS play a central role both upstream and downstream of the NF-κB and TNF-αpathways, which are the main mediators of the inflammatory response. A vicious circle is observed between ROS and these pathways. ROS are generated by the NOX system. Proteins modified by ROS could result in initiation of the auto-immune response to stimulate TNF-α and thus NOX. The dysregulation of these targets leads to the activation of several signaling pathways involved in cancer initiation.”

Are all of the effects of NSAIDs direct effects? Are any indirect effects, and if so, which pathways?

The observation of the reviewer is very relevant, at this effect we have added a paragraph and a Table 1 to describe the differential effects of NSAIDs:

Line 403:

“NSAIDs and the WNT/β-catenin pathway

NSAIDs downregulate the activity of COX-2 and thereby inhibit the synthesis of prostaglandins (PGE2) and then the WNT pathway [172]. Recent studies have observed that NSAIDS can have an anti-tumor effect, having revealed a chemo preventive effect against colon cancer [173–175]. The possible cellular pathway underlying the chemo preventive effect of NSAIDs involves the induction of cell-cycle arrest, apoptosis, and angiogenesis inhibition [174,175]

Several studies have also shown that NSAIDs can inhibit the canonical Wnt/β-catenin pathway [22]. Both aspirin and indomethacin downregulate the transcriptional activity of β-catenin/TCF-responsive genes [176]. NSAIDs reduce nuclear β-catenin levels and induce β-catenin degradation [177]. The NSAIDs, such as sulindac, exisulind and celecoxib, decrease β-catenin levels and then inhibit the transcriptional activity of the β-catenin/TCF/LEF complex [178]. Celecoxib directly inhibits cancer cell growth by downregulating the expression of the WNT/β-catenin pathway [179]and by inducing the degradation of the TCF7L2 [180]. Sulindac also inhibits the WNT/β-catenin pathway by downregulating nuclear β-catenin localization and β-catenin/TCF target gene transcription [181]. Colon cancer therapy with celecoxib is associated with an inhibition of the canonical WNT/β-catenin pathway [182,183]. Celecoxib inhibits the activity of the complex TCF/LEF and thus the activity of cyclin D1, suggesting that this compound inhibits the expression of WNT/β-catenin target genes [183]. Aspirin decreases glioma cell proliferation and invasion by inhibiting β-catenin/TCF transcription [184]. It also stops glioma cells cycle at the G0/G1 phase and inhibits invasion and tumor growth by downregulating β-catenin/TCF activity [184,185].”

And Line 560:

Table 1: Differential effects of NSAIDs in tumors

NSAIDs

Target

Target function

Interaction

References

Aspirin

PGE2

Immune system attenuation

Inhibition of PGE2 synthesis

[246]

Aspirin

Platelets

Reduction cell activity

Inhibition of COX

[247]

Aspirin

Genetic mutations

Tumorigenesis inhibition

Downregulation of gene mutation accumulation

[248]

Indomethacin

T-cell therapy

Tumorigenesis inhibition

Downregulation of cellular drug resistance

[247]

Aspirin

WNT pathway

Inhibition of cell proliferation and invasion

Inhibition of β-catenin accumulation

[184]

Aspirin

WNT pathway

Tumor suppressor

Inhibition of COX

[39]

Aspirin

WNT pathway

Arrest G0/G1 phase

Inhibition of β -catenin/TCF

[184,185]

Sulindac

WNT pathway

Inhibition of invasion and cell growth

Phosphorylation of Akt signaling

[51,52]

Aspirin and indomethacin

WNT pathway

Tumorigenesis inhibition

Inhibition of β -catenin and TCF/LEF

[176]

celecoxib

WNT pathway

Inhibition cancer cell growth

Inhibition of WNT

[179]

celecoxib

WNT pathway

Inhibition cancer cell growth

Inhibition of TCF/LEF

[180]

Sulindac

WNT pathway

Tumorigenesis inhibition

Inhibition of β-catenin accumulation

[181]

celecoxib

WNT pathway

Tumorigenesis inhibition

Inhibition of cyclin D1

[182,183]

PGE2: prostaglandin E2, COX: cyclooxygenase, TCF/LEF: T-cell factor/lymphoid enhancer-binding factor

References should also be updated, as the majority of the references are more than 5 years old.

We have corrected the references as demanded by the reviewer.

Reviewer 2 Report

This review paper is devoted to the potential role of nonsteroidal anti-inflammatory  drugs (NSAIDs) in inhibition of cancer growth and development. Particularly the differential effect of these drugs on canonical WNT/beta –catenin pathway and receptor PPAR gamma  is discussed. Also a hypothesis is given that administration of NSAIDs such as aspirin may reduce cancer incidence. In cancer there is observed aberrant  canonical WNT/beta –catenin pathway, which is upregulated. On the other hand  receptor PPARgamma  is down regulated, its activation inhibits  canonical WNT/beta –catenin pathway.  NSAIDs  acts as agonists of  PPARgamma receptors.  In this review the interplay between PPARgamma and WNT/beta-catenin pathways in cancer is broadly discussed. Important part of the review is also description of the relation between of circadian clock regulation and cancer,  and between circadian clock and inflammation and of the role of  WNT/beta –catenin pathway  and PPARgamma in this phenomenon. The last part of the review paper concerns molecular mechanisms of the action NSAIDs in cancer.

It is an interesting paper on the molecular mechanisms of action of NSAIDs in cancer. However some parts of the paper needs  language and other corrections. As example: in lines 20-21 (“In cancer process…..), line 84 (chronic initiation ?) line 119,  in  line 138 and 144-5 nearly the same statements,   160-161 –grammar, 171 – binds – grammar, line 344 – ROS gammat?  

Author Response

The authors thank the Reviewer for his criticisms and suggestions, which have improved the form and substance of our manuscript. We hope to have answered all the questions asked. 

The additions and changes requested are written in red in the manuscript.

This review paper is devoted to the potential role of nonsteroidal anti-inflammatory  drugs (NSAIDs) in inhibition of cancer growth and development. Particularly the differential effect of these drugs on canonical WNT/beta –catenin pathway and receptor PPAR gamma  is discussed. Also a hypothesis is given that administration of NSAIDs such as aspirin may reduce cancer incidence. In cancer there is observed aberrant  canonical WNT/beta –catenin pathway, which is upregulated. On the other hand  receptor PPARgamma  is down regulated, its activation inhibits  canonical WNT/beta –catenin pathway.  NSAIDs  acts as agonists of  PPARgamma receptors.  In this review the interplay between PPARgamma and WNT/beta-catenin pathways in cancer is broadly discussed. Important part of the review is also description of the relation between of circadian clock regulation and cancer,  and between circadian clock and inflammation and of the role of  WNT/beta –catenin pathway  and PPARgamma in this phenomenon. The last part of the review paper concerns molecular mechanisms of the action NSAIDs in cancer.

It is an interesting paper on the molecular mechanisms of action of NSAIDs in cancer. However some parts of the paper needs  language and other corrections. As example: in lines 20-21 (“In cancer process…..), line 84 (chronic initiation ?) line 119,  in  line 138 and 144-5 nearly the same statements,   160-161 –grammar, 171 – binds – grammar, line 344 – ROS gammat?  

The authors thank the reviewer, we have corrected our manuscript by an English native corrector.

Reviewer 3 Report

Vallee and group have reviewed the current literature of the Wnt/beta-catenin pathway and its targeting using NSAIDs in cancer. Authors summarized the role of chronic inflammation and oxidative stress in cancer as well as their interaction. Next, the relationship between the Wnt pathway, chronic inflammation and oxidative stress in cancer is summarized in detail. Further, the involvement of Circadian clock involvement in the cancer is reviewed. The next section, the author summarized NSAIDs as lead drugs for the treatment of cancer. The topic is worth reviewing and hence will be interesting to researchers. I recommend this article for acceptance after revision.

1.    The current treatment regime for cancer (radiotherapy, chemotherapy, surgery) can be discussed in the introduction. The current problems and why NSAIDs will be beneficial for the treatment of cancer can be included. 

2.     Make a table summarizing finding with different NSAIDs and Wnt pathway and cancer

3.     I feel that some sections are under-references such as NSAIDs inhibit COX-2, NSAIDs and Akt pathway, NSAIDs and oxidative stress in cancer.

4.    Please discuss strategies to inhibit the Wnt-beta-catenin pathway briefly or make a table such as Wnt inhibitor OMP-54F28, Frizzled antibodies, Tankyrase inhibitors, CBP inhibitors, PORCN inhibitor and COX inhibitors inhibiting beta-catenin accumulation. I think repurposing NSAIDs as COX inhibitors will be the best strategy because you can repurpose a lot of drugs.

Author Response

The authors thank the Reviewer for his criticisms and suggestions, which have improved the form and substance of our manuscript. We hope to have answered all the questions asked. 

The additions and changes requested are written in red in the manuscript.

Vallee and group have reviewed the current literature of the Wnt/beta-catenin pathway and its targeting using NSAIDs in cancer. Authors summarized the role of chronic inflammation and oxidative stress in cancer as well as their interaction. Next, the relationship between the Wnt pathway, chronic inflammation and oxidative stress in cancer is summarized in detail. Further, the involvement of Circadian clock involvement in the cancer is reviewed. The next section, the author summarized NSAIDs as lead drugs for the treatment of cancer. The topic is worth reviewing and hence will be interesting to researchers. I recommend this article for acceptance after revision.

1.        The current treatment regime for cancer (radiotherapy, chemotherapy, surgery) can be discussed in the introduction. The current problems and why NSAIDs will be beneficial for the treatment of cancer can be included. 

According to the reviewer, we have re-written the introduction

“The complex process of cancer can be defined in terms of three stages: initiation, promotion and progression [1–3]. Many cancers are initiated by chronic inflammation, involving numerous physical, chemical and biological determinants [4]. Several studies have examined the relationship between chronic inflammation and cancer [5,6]and indeed have highlighted the promising role of anti-inflammatory treatments for cancer [7]. Chronic inflammation is responsible for the different stages observed in cancers, such as invasion, angiogenesis, proliferation and metastasis [8–10].

In parallel, oxidative stress causes DNA damage in cancers [11]. In the past few years, the combined effect of oxidative stress and chronic inflammation has been the subject of a number of studies [12]. Reactive oxygen species production (ROS) is increased by the activation of inflammatory factors [13–15]and thus also participates in the processes of invasion, proliferation, angiogenesis and then metastasis [16]. The canonical WNT/β-catenin pathway controls several other pathways involved in cancer development and tissue homeostasis. This pathway is regulated from transcription-level regulations to post-transcriptional modifications. In cancers, an aberrant WNT/β-catenin pathway is generally observed and leads to oxidative stress and inflammation [12,17,18]

Numerous epidemiological studies have shown that non-steroidal anti-inflammatory drugs (NSAIDs) could have a positive effect in both the prevention of cancer and tumor therapy. Moreover, the regular administration of aspirin, a NSAID, has been found to be correlated with a reduction in cancer incidence [19]. A regular therapy of more than 75mg/day of aspirin reduces the incidence of several cancers and tumor metastases, leading to an improvement in survival rates [20]. Regular use of NSAIDs is associated with a reduced incidence of several cancers, such as breast cancer, lung cancer and gliomas [21,22]. Recent data have shown that the use of aspirin is associated with a reduction in the incidence of death from cancer, as well as in metastatic spread [19,20,23]. Anti-inflammatory drugs are commonly used in clinical practice due to their analgesic, anti-inflammatory and antipyretic effects. Furthermore, NSAIDs are often used in conjunction with other drugs in treating a number of diseases. Numerous hypotheses have postulated that NSAIDs could decrease tumor growth by acting on both chronic inflammation and oxidative stress [24]. Anti-inflammatory drugs could be used to target the chronic inflammatory microenvironment of tumors. It is well known that the human body is capable of self-healing after a short-term inflammatory response, but a long-term chronic inflammation could lead to initiation of the cancer process. Many studies have shown that inflammatory factors, including interleukins, TNF-α, Nuclear factor-ϰB (NF-ϰB) and ROS production-induced inflammation, infiltrate the inflammatory microenvironment, leading to DNA damage and ultimately initiation of the cancer process [25,26]

NSAIDS act as peroxisome proliferator-activated receptor gamma (PPARγ) agonists and could thus downregulate the aberrant WNT/β-catenin pathway in cancers [22]. PPARγ agonists offer an interesting therapeutic solution in cancers by acting on both oxidative stress and inflammation [27,28]. Indeed, in several tissues, canonical WNT/β-catenin pathway activation induces inactivation of PPARγ, while PPARγ activation induces inhibition of the canonical WNT/β-catenin pathway. In most cancers, the canonical WNT/β-catenin pathway is increased while PPARγ is downregulated [12]. In parallel, dysregulation of circadian rhythms (CRs) has been observed in cancers [29]. This dysfunction leads to upregulation of the canonical WNT/β-catenin pathway, which contributes to the cancer process. PPARγ can control CRs by regulating many key circadian genes, like Bmal1 (brain and muscle aryl-hydrocarbon receptor nuclear translocator-like 1) [30], and can directly target the WNT pathway [31]. Numerous evidence points to the anti-cancer benefits of NSAIDs, even if these benefits remain unclear and poorly understood. Nevertheless, data from experiments suggest a potential role for NSAIDs in the treatment of cancer through the regulation of the WNT/β-catenin pathway [32].

This review focuses on the interest of using NSAIDs in cancer therapy through their capacity to regulate the aberrant canonical WNT/b-catenin pathway and PPARγ, two systems that respond in an opposite manner.”

2.        Make a table summarizing finding with different NSAIDs and Wnt pathway and cancer

This is a relevant observation of the reviewer, at this effect we have made a Table describing the different NSAIDs and their actions on the WNT. Line 560

Table 1: Differential effects of NSAIDs in tumors

NSAIDs

Target

Target function

Interaction

References

Aspirin

PGE2

Immune system attenuation

Inhibition of PGE2 synthesis

[246]

Aspirin

Platelets

Reduction cell activity

Inhibition of COX

[247]

Aspirin

Genetic mutations

Tumorigenesis inhibition

Downregulation of gene mutation accumulation

[248]

Indomethacin

T-cell therapy

Tumorigenesis inhibition

Downregulation of cellular drug resistance

[247]

Aspirin

WNT pathway

Inhibition of cell proliferation and invasion

Inhibition of β-catenin accumulation

[184]

Aspirin

WNT pathway

Tumor suppressor

Inhibition of COX

[39]

Aspirin

WNT pathway

Arrest G0/G1 phase

Inhibition of β -catenin/TCF

[184,185]

Sulindac

WNT pathway

Inhibition of invasion and cell growth

Phosphorylation of Akt signaling

[51,52]

Aspirin and indomethacin

WNT pathway

Tumorigenesis inhibition

Inhibition of β -catenin and TCF/LEF

[176]

celecoxib

WNT pathway

Inhibition cancer cell growth

Inhibition of WNT

[179]

celecoxib

WNT pathway

Inhibition cancer cell growth

Inhibition of TCF/LEF

[180]

Sulindac

WNT pathway

Tumorigenesis inhibition

Inhibition of β-catenin accumulation

[181]

celecoxib

WNT pathway

Tumorigenesis inhibition

Inhibition of cyclin D1

[182,183]

PGE2: prostaglandin E2, COX: cyclooxygenase, TCF/LEF: T-cell factor/lymphoid enhancer-binding factor

3.        I feel that some sections are under-references such as NSAIDs inhibit COX-2, NSAIDs and Akt pathway, NSAIDs and oxidative stress in cancer.

We have corrected theses under-references by developing these paragraphs:

Line 122:

“NSAIDs and cancer prevention

A strategy for reducing cancer risks could involve the use of NSAIDs (such as aspirin, naproxen or ibuprofen) [20]. Several studies have shown a correlation between the long-term use of aspirin and a reduction in both the incidence and mortality of cancers, a reduction that can vary from 20 to 75% [20]. The most marked effects have been observed in colorectal cancers, stomach cancers and esophageal adenocarcinoma, while less marked effects have been found in lung, prostate and breast cancers [20]. In contrast, little benefit has been observed in pancreatic and endometrial cancers [20]. Numerous molecular mechanisms could explain the link between NSAIDs and cancer prevention, such as COX inhibition, immune response, PI3K/Akt pathway downregulation, pro-inflammatory response and decreased glycolytic signaling in tumor cells [22,33]. COX inhibition is associated with the reduction in inflammatory mediators including prostaglandins [34]. The activation of COX in the cancer process leads to the expression of prostaglandin E3 (PGE2), which induces angiogenesis, tumor growth and metastasis [35]. Furthermore, PGE2 stimulates several signaling pathways, such as the PI3K/Akt and NF-ϰB, which induce tumorigenesis [35]. Finally, recent studies have shown that NSAIDs could also act on other signaling pathways, such as iNOS, TNF-alpha and interleukins [33].

NSAIDs lead to cancer cell apoptosis

The use of NSAID (aspirin) in ovarian cancer cells decreases Bcl-2 expression and increases Bax gene expression [24]. The role of Bcl-2 in cancer is to inhibit apoptosis by changing mitochondria thiol, affecting mitochondria membrane permeability and translocating to the mitochondria membrane the apoptotic protein precursor Apaf-1 to inhibit the role of the latter. The NSAID sulindac inhibits the expression of both the protein Bcl-XL and the Bcl-XL antagonist of Bax to induce activation of caspase cascade, which stimulates the apoptosis process [36]. Aspirin can change mitochondrial permeability to downregulate Bcl-2 expression, block ATP synthesis and release cytochrome C, which triggers apoptosis [37]. The NSAID celecoxib activates the p53up-regulated modulator of apoptosis (PUMA) to increase p53 expression and thus initiate apoptosis [38]

 NSAIDs inhibit COX-2

NSAIDs are known to act as a tumor suppressor through the inhibition of COX [39]. COX presents three subtypes: COX-1, COX-2, and COX-3 (in nervous system)[40,41]. COX-1 catalyzes the production of prostaglandins (PGs) to maintain physiological functions. COX-2, a membrane-bound protein, is not expressed in normal cells but over-expressed in inflammation and tumors [42]. PG over-expression is induced by COX-2 catalysis during inflammation that leads to neovascularization to provide nutrition for tumor proliferation [43]. In parallel, in tumors, COX-2 upregulates Bcl-2 expression to initiate the anti-apoptotic process [44], while it modulates MMP-2 expression, which induces tumor invasion and metastasis [45]. Celecoxib, a NSAID and a COX-2 enzyme inhibitor, can downregulate tumor proliferation and can induce apoptosis in a variety of tumor cells [24]. By directly blocking COX-2 expression, NSAIDS could prevent cancer initiation [24].

NSAIDs and the Akt pathway

Colorectal cancer presents epidermal growth factor receptor (EGFR) overexpression and the use of aspirin can downregulate EGFR [46]. EGFR is involved in several pathophysiological responses in cancer, such as migration, proliferation and invasion [47,48]. The Akt pathway is activated by EGFR [49]. An aberrant WNT/β-catenin pathway stimulates EGFR expression [50]. Furthermore, NSAIDs dephosphorylate Akt signaling and decrease MMP-2 gene expression to inhibit invasion and cell growth [51,52].”

4.        Please discuss strategies to inhibit the Wnt-beta-catenin pathway briefly or make a table such as Wnt inhibitor OMP-54F28, Frizzled antibodies, Tankyrase inhibitors, CBP inhibitors, PORCN inhibitor and COX inhibitors inhibiting beta-catenin accumulation. I think repurposing NSAIDs as COX inhibitors will be the best strategy because you can repurpose a lot of drugs. 

The reviewer is right, we have added a paragraph. Line 338

WNT pathway is the target of several inhibitors in therapeutic strategies to counteract tumorigenesis, such as OMP-54F28 [132–134], Frizzled antibodies [135], tankyrase inhibitors [136], CBP inhibitors [137], and PORCN inhibitors [138].

Round 2

Reviewer 1 Report

The revised manuscript is substantially improved. A few minor issues still need to be corrected:

Line 115 NSAIDs are not a tumor suppressor, which is defined as a gene or protein “that protects a cell from one step on the path to cancer.”

Line 202 should be eliminated.

Lines 222-227 are 2 sentence paragraphs and should be combined or edited to fit in with the ideas of other more complete paragraphs.

Lines 314-316 should be incorporated into the paragraph above this sentence.

Line 485 “pain reported a twice-daily flurbiprofen schedule” should be revised so the reader understands what Kowanko et al reported.

Author Response

The authors thank the Reviewer for his criticism and suggestions, which have improved the form and substance of our manuscript. We hope to have answered all the questions asked. 

The additions and changes requested are written in blue in the manuscript.

The revised manuscript is substantially improved. A few minor issues still need to be corrected:

Line 115 NSAIDs are not a tumor suppressor, which is defined as a gene or protein “that protects a cell from one step on the path to cancer.”

The reviewer is right, according to his relevant remark we have change our sentence to

Line 115, “NSAIDs are known to protect cells from one step on the path to cancer through the inhibition of COX”

Line 202 should be eliminated.

According to the reviewer, we have suppressed this sentence: “The migration and invasion of cancer cells are facilitated by inflammatory factors [91].”

Lines 222-227 are 2 sentence paragraphs and should be combined or edited to fit in with the ideas of other more complete paragraphs.

According to the relevant observation of the reviewer, we have change the sentences:

Line 222, “Cell damage from oxidation and nitration of macromolecules enhances ROS production by activation of the NADPH oxidase (NOX) enzyme. This phenomenon leads to the reduction of the transfer of electrons through the mitochondrial membrane to reduce the molecular oxidative metabolism.”

Lines 314-316 should be incorporated into the paragraph above this sentence.

The reviewer is right, we have included this sentence Line 270.

Line 485 “pain reported a twice-daily flurbiprofen schedule” should be revised so the reader understands what Kowanko et al reported.

The reviewer is right. We have changed this sentence: 

Line 480: “Nevertheless, Kowanko et al. observed that pain reported in rheumatoid arthritis after a twice a day therapy by flurbiprofen may be more effective than four times daily flurbiprogen, and that regimen without an evening dose was the least effective of three twice-daily treatments tested. Moreover, their results suggested that morning stiffness in rheumatoid arthritis was not only the result of nocturnal inactivity but also a response to an appropriately timed medication given to decrease inflammation or immune response [223].”